# Comprehensive Transcriptomic Comparison between Porcine CD8^−^ and CD8^+^ Gamma Delta T Cells Revealed Distinct Immune Phenotype

**DOI:** 10.3390/ani11082165

**Published:** 2021-07-22

**Authors:** Sangwook Kim, Byeonghwi Lim, Sameer-ul-Salam Mattoo, Eun-Young Oh, Chang-Gi Jeong, Won-Il Kim, Kyung-Tai Lee, Sang-Myeong Lee, Jun-Mo Kim

**Affiliations:** 1Functional Genomics & Bioinformatics Lab, Department of Animal Science and Technology, Chung-Ang University, Anseong 17546, Korea; genemap4077@cau.ac.kr (S.K.); hwi1208@cau.ac.kr (B.L.); 2Division of Biotechnology, College of Environmental & Biosource Science, Jeonbuk National University, Iksan 54596, Korea; drsameerulsalam@gmail.com; 3College of Veterinary Medicine, Chungbuk National University, Cheongju 28644, Korea; rwae00@naver.com; 4College of Veterinary Medicine, Jeonbuk National University, Iksan 54596, Korea; jcg0102@gmail.com (C.-G.J.); kwi0621@jbnu.ac.kr (W.-I.K.); 5Animal Genomics and Bioinformatics Division, National Institute of Animal Science, RDA, Wanju 55365, Korea; leekt@korea.kr

**Keywords:** gamma delta T cell, CD8, RNA-seq, immune mechanisms, pig

## Abstract

**Simple Summary:**

This study was conducted to comprehensively understand the functional mechanisms of CD8^+/−^ porcine gamma delta (γδ) T cells related to the immune system using RNA-sequencing technology. In total, 646 upregulated and 561 downregulated differentially expressed genes (DEGs) for CD8^+^ were identified and functional annotation was performed. A cytokine–cytokine receptor interaction and T cell receptor (TCR) signaling pathway were enriched in the upregulated DEG group, whereas the B cell receptor signaling pathway was enriched in the downregulated DEG group. Chemokine-related genes (*CXCR3*, *CCR5*, *CCL4*, *CCL5*), interferon gamma (*IFNG*), and CD40 ligand (*CD40LG*) identified in the cytokine–cytokine receptor interaction and TCR signaling pathway may affect the inter-regulation of immune signaling. Our results are expected to contribute to the understanding of mechanisms of porcine γδ T cells.

**Abstract:**

We aimed to comprehensively understand the functional mechanisms of immunity, especially of the CD8^+/−^ subsets of gamma delta (γδ) T cells, using an RNA-sequencing analysis. Herein, γδ T cells were obtained from bronchial lymph node tissues of 38-day-old (after weaning 10-day: D10) and 56-day-old (after weaning 28-day: D28) weaned pigs and sorted into CD8^+^ and CD8^−^ groups. Differentially expressed genes (DEGs) were identified based on the CD8 groups at D10 and D28 time points. We confirmed 1699 DEGs between D10 CD8^+^ versus D10 CD8^−^ groups and 1784 DEGs between D28 CD8^+^ versus D28 CD8^−^ groups; 646 upregulated and 561 downregulated DEGs were common. The common upregulated DEGs were enriched in the cytokine–cytokine receptor interaction and T cell receptor (TCR) signaling pathway, and the common downregulated DEGs were enriched in the B cell receptor signaling pathway. Further, chemokine-related genes, interferon gamma, and CD40 ligand were involved in the cytokine–cytokine receptor interaction and TCR signaling pathway, which are associated with inter-regulation in immunity. We expect our results to form the basic data required for understanding the mechanisms of γδ T cells in pigs; however, further studies are required in order to reveal the dynamic changes in γδ T cells under pathogenic infections, such as those by viruses.

## 1. Introduction

Research on the porcine immune system has considerably increased because of the importance of swine models in agricultural and biomedical fields. The interaction between a pathogen and the immune system plays an important role in agricultural research, and a comprehensive understanding of the functions of the porcine immune system is required in order to identify the associated immune mechanisms and define the immune interactions. In particular, pigs after weaning show impaired growth and an increased incidence of several diseases, including diarrhea, due to severe environmental alterations [1,2].

The adaptive immune system, based on the functions of the T and B lymphocytes, is one of the primary research areas. The T and B lymphocytes stimulated by the interaction of cells from the innate immune system, such as dendritic cells and monocytes, show a pathogen-specific reaction and further retain this immunological memory. In addition, the activation of the B lymphocytes is likely to be detected by antibodies, whereas the monitoring of the antigen-specific T cell response requires detailed knowledge and understanding of the phenotypes of each T cell subpopulation and T lymphocyte participating in the immune reaction [3].

The porcine T cells are classified into two lineages based on the presence of either an alpha beta (αβ) T cell receptor (TCR) or a gamma delta (γδ) TCR. The porcine αβ T cells recognize antigens in a major histocompatibility complex (MHC)-restricted manner, whereas the γδ T cells recognize them by an MHC non-restricted approach [4]. The porcine γδ T cells are a subset of T cells that have a TCR complex independent of the αβ T cells. Their various potential functions include protecting against extra- and intra-cellular pathogens, performing tumor surveillance, modulating innate and adaptive immune reactions, performing tissue healing, maintaining epithelial cells, reacting against various viral infections, and regulating the physiological organ functions [5,6,7,8].

The frequency and profile of the γδ T cells are different in different species. For example, the γδ T cells in humans and rodents represent only a minor fraction (approximately 1–5% in blood) in the spleen, lymph nodes, and peripheral blood, and lead to tissue tropism on the surface of the epithelium [9], whereas in swine, the γδ T cells are rich in blood, can occupy as much as 85% of the total lymphocytes, and are not restricted to the epithelium, making the swine a member of the “γδ- high species” group, where the γδ T cells account for a high ratio of the total T cells in some swine breeds [10,11].

Based on the expression of the co-receptors, the porcine γδ T cells are subdivided into CD8^−^ and CD8^+^ subsets. The CD8^+^ T cell subset is likely to have potential cytotoxic activity [4]. The current research on the CD8^+^ γδ T cells, a major component of the porcine immune system, has focused only on identifying the immunological importance by investigating their classification and profiles, and understanding their functional mechanisms in livestock, such as swine. In addition, several studies on viral diseases in swine have investigated the increase and activity of CD8^+^ T cells by infecting swine with pathogens [12,13,14] and have revealed that, in particular, the damage induced by viral diseases in the early stage of weaning pigs (up to approximately 3–4 weeks) is critical even after gaining passive immunity from sow [15,16]. However, the functions of the CD8^+^ γδ T cells in non-infected animals have not been comprehensively understood yet.

The purpose of this study was to comprehensively understand the immunological functional mechanisms of CD8^+^ γδ and CD8^−^ γδ T cells by performing bioinformatics analysis. Since the respiratory tract is one of the major routes of viral or bacterial infection, we performed RNA-sequencing (RNA-seq) analysis of the γδ T cells acquired from bronchial lymph nodes from 10 pigs that were 10 (*n* = 5) and 28 (*n* = 5) days after weaning (the early stage of weaning).

## 2. Materials and Methods

### 2.1. Experimental Animals

Ten 4-week-old piglets were purchased from a porcine reproductive and respiratory syndrome virus (PRRSV)-free farm. After acclimation, five pigs were euthanized at 38 days (after weaning 10-day: D10) of age and the remaining five were euthanized at 56 days (after weaning 28-day: D28) of age. Euthanasia was performed by electrocution after an intramuscular injection of 2 mL azaperone (40 mg/mL; Stress Guard^®^, Dong Bang Inc., Seoul, Korea). Feed and water were provided ad libitum to all pigs.

### 2.2. Sorting of the γδ T Cell Subsets

Bronchial lymph nodes were collected from the D10 and D28 pigs and were stored in ice-cold phosphate-buffered saline (PBS). Single-cell suspensions of the lymph nodes were prepared according to a previous study [8]. Briefly, the lymph nodes were gently crushed and passed through a 40 μm cell strainer to prepare single-cell suspensions. The suspensions were then washed twice with PBS, followed by RBC lysis using an RBC lysis buffer (eBioscience^TM^, San Diego, CA, USA). Eventually, the cells were re-suspended in a fluorescent antibody cell sorting (FACS) buffer containing 3% heat-inactivated fetal bovine serum (Gibco) and 0.02% sodium azide in PBS. The cells were stained with a δ chain-specific TCR1 antibody (anti-swine TCR1 δ mAb, clone: PGBL22A, cat: WS0621S-10, Kingfisher Biotech) followed by APC-conjugated rat anti-mouse IgG1 antibody (clone RMG1-1, cat: 406610, BioLegend). Next, the samples were stained with CD3: FITC (clone: PPT3, cat: MCA5951F, Bio-Rad) and CD8a: PE (clone: 76-2-11, cat: 559584, BD-Biosciences) antibodies. At each staining step, the samples were incubated on ice for 30 min in dark and then subsequently washed twice with ice-cold FACS buffer. After staining, the cells were re-suspended in the FACS buffer and sorted by FACS Aria; based on the expression of CD8, the cells were sorted into two different tubes as CD3^+^TCR1δ^+^ CD8^+^ and CD3^+^ TCR1δ^+^CD8^−^ cells. The purity of each T cell subset was determined using flow cytometry and a purity of >99% was achieved (Appendix A). Finally, one million cells were stored in liquid nitrogen and subsequently used for RNA-seq analysis.

### 2.3. Performing RNA Quantification and Determining RNA Quality

RNA was extracted from 20 types of cells, according to the sorted γδ T cell types (CD8^+^ or CD8^−^) and time points (10th or 28th day) using TRIzol™ (Invitrogen, Waltham, MA, USA) as per the manufacturer’s protocol. Briefly, the cell samples were homogenized with 2.0 mL of TRIzol and chloroform, and then precipitated using isopropanol (Junsei Chemical Ltd., Tokyo, Chou-ku, Japan). The extracted RNA samples were maintained at −80 °C. Contamination of the genomic DNA was removed by treating 25 μg of RNA from each sample with RNase-free DNase set (QIAGEN, Hilden, Germany) and RNA purification was performed using an RNeasy mini kit (QIAGEN) by following the manufacturer’s instructions. The quality of the RNA was measured using RNA 6000 Nano LabChip from Bioanalyzer 2100 (Agilent Technologies, Santa Clara, CA, USA) and automated capillary gel electrophoresis was performed as per the manufacturer’s protocol.

### 2.4. Library Preparation for High-Throughput Transcriptomic Sequencing

A library of template molecules suitable for cluster generation was prepared using the extracted mRNA. The Illumina TruSeq RNA Sample Prep Kit (Illumina, Inc., San Diego, CA, USA) was used to prepare the Ribo-Zero RNA-Seq libraries according to the manufacturer’s protocol. Then, rRNA was removed from the total RNA with the Ribo-Zero rRNA Removal Kit (Epicentre, Madison, WI, USA). The total RNA without rRNA was purified, fragmented, and primed for cDNA synthesis. According to the manufacturer’s instructions for the automated capillary gel electrophoresis, 0.1–4.0 μg of total RNA from each sample was used for the cDNA library preparation and sequencing. Agilent Technologies Human UHR total RNA was used as a positive control sample. Following the standard protocol provided by Illumina, five technical replicates were used to prepare the library. The sequencing was performed on one lane after mixing libraries with different indexes into one group. The library was sequenced by 100 paired-end sequencing using Illumina HiSeq 2000 high-power sequencing tools, producing a total of 1.2 billion paired-end sequence reads, and resulting in an average of 126.5 million reads per sample (Appendix A).

### 2.5. Quality Control and Reads Mapping to the Reference Genome

For the quality filtering strategy, a quality check of the raw reads data was performed using the FastQC v0.11.7 software for each sample [17]. The adapter and single-end reads, based on the quality outcomes, were identified and removed using Trimmomatic v0.38 tool [18]. Subsequently, the trimmed reads were re-checked using FastQC [19] and were mapped to the reference genome (*Sus scrofa* 11.1, GCA_000003025.6) from the Ensembl genome browser (http://www.ensembl.org/Sus_scrofa/ accessed on 1 October 2020) using the default option of HISAT2 v2.1.0 program [19]. Raw counts of genes for each library were calculated using the *Sus scrofa* GTF v97 (Ensembl) file, a genomic annotation reference file, using the “featureCounts” function from the “Subread” package v.1.6.3 [20].

### 2.6. Differentially Expressed Genes (DEGs) Analysis

The DEG analyses were performed using the “edgeR” package v3.26.5 of Bioconductor for the obtained raw counts [21]. Normalization of the raw counts was performed using the trimmed mean of M-values (TMM) method [22] and the dispersion parameter was estimated and applied using the Cox–Reid method. A negative binomial linear model was used to determine the DEGs from four case combinations based on the γδ T cell type (CD8^+^ or CD8^−^) and number of days (10 or 28 days after weaning, D10 or D 28): (i) D10 CD8^+^ vs. D28 CD8^−^, (ii) D28 CD8^+^ vs. D28 CD8^−^, (iii) D10 CD8^+^ vs. D10 CD8^−^, and (iv) D28 CD8^+^ vs. D10 CD8^+^. The cutoff criteria included an adjusted *p*-value using the Benjamini–Hochberg correction with a false discovery rate (FDR) < 0.05 and absolute log2 fold change (FC) ≥ 1 [23]. To check for similarity among the samples, multidimensional scaling (MDS) was calculated using the “limma” package from R [24], and visualization was performed using the “ggplot2” package [25]. Further analyses were performed after filtering outliner based on MDS result.

### 2.7. Gene Ontology and Kyoto Encyclopedia of Genes and Genomes Enrichment Analyses of the DEGs

The biological processes (BPs), cellular components (CCs), and molecular functions (MFs) of the gene ontology (GO) terms and the Kyoto Encyclopedia of Genes and Genomes (KEGG) pathways were analyzed using the Database for Annotation, Visualization, and Integrated Discovery (DAVID) v6.8, using the selected DEG sets [26]. The GO annotations were filtered using the DIRECT option and then subjected to enrichment analysis with cutoff criteria of *p*-value < 0.1 and count ≥ 2. Subsequently, the treemaps of the enriched GO terms were visualized using the REVIGO tool [27]. The KEGG annotations were enriched using identical cutoff criteria and expressed as −log10 *p*-values and fold enrichment values. Before the enrichment analysis, the data were annotated on the *Sus scrofa*.

### 2.8. Gene Set Enrichment Analysis, Pathway Profiling, and Protein—Protein Interaction Network Analysis

Gene set enrichment analyses (GSEA) were performed using the GSEA v4.0.2 software with CD8^+^ samples as a case and CD8^−^ samples as a control. The analyses were conducted using the gene-ranking method based on the gene sets in the KEGG database in order to determine the enrichment scores and significant differences [28]. All analyses were performed using the log2-normalized TMM counts of the samples. GSEA results were visualized as a bubble plot with significant pathways (*p*-value < 0.05). Additionally, the core enriched genes of the main pathways showing high normalized enrichment scores were visualized via heatmaps. Next, the pathway profiling of the corresponding gene products (proteins) of the selected representative main KEGG pathways (as determined via the DEGs and GSEA) were confirmed using the “clusterProfiler” package in R software [29]. From the genes corresponding to each protein, the genes showing the maximum changes were used as the representative values. Finally, the protein–protein interaction (PPI) network of each pathway was investigated to identify significant genes using the *Homo sapiens* database of STRING v11.0 [30].

## 3. Results

### 3.1. Quality Control Analysis and RNA-Seq Data

The percentages of GC, AT, Q20, and Q30 in the cleaned reads were calculated and are shown in Appendix A for all 18 samples (D10 CD8^−^: *n* = 5, D10 CD8^+^: *n* = 4, D28 CD8^−^: *n* = 4, D28 CD8^+^: *n* = 5). The mean read value for the total RNA from all γδ T cells was 126,485,253 and those for D10 CD8^−^, D10 CD8^+^, D28 CD8^−^, and D28 CD8^+^ samples were 135,330,227, 129,905,226, 125,532,421, and 115,666,518, respectively. In addition, the frequency of GC was 56.25% and the means of Q20 (%) and Q30 (%) were 96.78% and 92.15%, respectively, demonstrating a high quality.

### 3.2. Identification of the DEGs

The MDS analysis using the processed transcriptomic data showed that all the γδ T cell samples were clustered clearly by type (CD8^+^/CD8^−^) and days (D10/D28; Figure 1A). Due to the fact that 2 samples were filtered out as outliers, analyses were conducted with 18 samples thereafter.

Based on the cell type, a total of 1699 DEGs were identified between the D10 CD8^+^ and D10 CD8^−^ groups (Appendix A) and 1784 DEGs between the D28 CD8^+^ and D28 CD8^−^ groups (Appendix A and Figure 1B). Based on the age (D10 and D28), 492 DEGs were identified between the D28 CD8^−^ and D10 CD8^−^ groups and 490 DEGs between the D28 CD8^+^ and D10 CD8^+^ groups (FDR < 0.05, log2 FC ≥ 1; Appendix A). Next, we used a Venn diagram to comprehensively investigate the common DEGs between the samples with and without CD8, regardless of the number of days, and found that 1207 DEGs were common (Figure 1C), including 646 and 561 DEGs that were upregulated and downregulated, respectively, in the CD8^+^ samples rather than in the CD8^−^ samples for all of the days (D10 and D28; Appendix A).

Using the same approach, we observed that for the cell type-specific samples with different day-old pigs, the Venn diagram showed that the CD8^−^ specific and CD8^+^ specific DEGs were 307 and 305, respectively (Figure 1D).

### 3.3. Differences in the Functional Annotations

To investigate the differences in biological functions depending on the presence of CD8^+^ in γδ T cells, KEGG- (Figure 2A) and GO-based (Figure 2B) functional enrichment analyses were performed using the common 646 upregulated and 561 downregulated DEGs for the D10 CD8^+^ vs. D10 CD8^−^ and D28 CD8^+^ vs. D28 CD8^−^ groups. For the upregulated DEGs, the KEGG-enriched pathways included a cytokine–cytokine receptor interaction, TCR signaling pathway, natural killer cell-mediated cytotoxicity, and antigen processing and presentation, whereas for the downregulated DEGs, a cytokine–cytokine receptor interaction and B cell receptor (BCR) signaling pathway were enriched. Most of these pathways were biological and related to immune reactions, indicating that although the upregulated and downregulated genes had common biological reactions related to immunity, there was a major difference, wherein the KEGG-enriched pathways for the former included the TCR signaling pathway and those for the latter included the BCR signaling pathway. The treemaps indicated another difference, wherein the former group showed that the significantly enriched BP terms were mostly related to immune response, and for the latter, they were enriched in the BCR signaling pathway (Figure 2B). These enriched results were true in both the KEGG and GO analyses.

Additionally, the DEGs for the D28 CD8^−^ vs. D10 CD8^−^ and D28 CD8^+^ vs. D10 CD8^+^ groups were compared in order to identify the differences in functions according to the number of days, and the functional analysis using CD8^−^ specific DEGs (*n* = 307) and CD8^+^ specific DEGs (*n* = 305) revealed many terms related to immunity (Appendix A).

### 3.4. GSEA, Pathway Profiling, and PPI Network Analysis

For the validation of the functional enrichment analysis according to the presence of CD8^+^ in the γδ T cells, KEGG-based GSEA was performed using CD8^+^ samples as a case and CD8^−^ samples as a control. We observed significant enrichment of many reactions related to the immune system, and this was consistent with the results of the KEGG enrichment analysis (Figure 3A). Among the pathways that were significantly enriched in both the KEGG enrichment analyses, three important immunity-related pathways were selected: two pathways (cytokine–cytokine receptor interaction and TCR signaling pathway) from the upregulated DEGs and one pathway (BCR signaling pathway) from the downregulated DEGs. The selected pathways were visualized via heatmaps that displayed their expression level on the basis of the core enriched genes. Among the 52 core enriched genes identified by GSEA in the cytokine–cytokine receptor interaction, 25 genes were included in the upregulated DEG group (Figure 3B). The proteins (genes) identified by the pathway profiling analysis were: CCL3L1 (*CCL4*), CCL4 (*CCL4*), CCL4L1 (*CCL4*), CCL4L2 (*CCL4*), CCL5 (*CCL5*), CX3CR1 (*CX3CR1*), XCL1 (*XCL1*), CCL3 (*CCL3L1*), CXCR3 (*CXCR3*), IFNG (*IFNG*), IL18RAP (*IL18RAP*), CCR5 (*CCR5*), CCL16 (*CCL16*), GHR (*GHR*), IL2RB (*IL2RB*), IL12RB2 (*IL12RB2*), CSF1 (*CSF1*), FASLG (*FASLG*), CXCR6 (*CXCR6*), RANK (*TNFRSF11A*), CD40L (*CD40LG*), FAS (*FAS*), BAFF (*TNFSF13B*), TNF (*TNF*), CCR9 (*CCR9*), TGFBR2 (*TGFBR2*), 4-1BB (*TNFRSF9*), and CD27 (*CD27*; Figure 3C). In addition, among the 19 core enriched genes identified by GSEA from the TCR signaling pathway, 14 genes were included in the upregulated DEG group. The proteins (genes) identified by pathway profiling analysis were: IFN-γ (*IFNG*), CD4/8 (*CD8A*, *CD8B*), LCK (*LCK*), PAK (*PAK6*), CD28 (*CD28*), CBL (*CBLB*), CD40L (*CD40LG*), TNFα (*TNF*), PI3K (*PIK3R1*), ICOS (*ICOS*), NCK (*NCK1*), ITK (*TEC*), and p38 (*MAPK13*; Figure 3C). From the BCR signaling pathway, six genes from 22 core enriched genes were included in the upregulated DEG group: VAV (*VAV2*), BCAP (*PIK3AP1*), SYK (*SYK*), CD19 (*CD19*), Igβ (*CD79B*), and IκB (*NFKBIA*; Figure 3C).

PPI networks using the abovementioned significant genes from the three selected main pathways were constructed and, based on them, the proteins (genes) were shown to be linked organically for each pathway (Figure 4). In the cytokine–cytokine receptor interaction network, CXCR3, IFNG, CCR5, TNF, CCL4, CCL5, CD40L, and CD27 proteins from the 25 total proteins occupied most (≥15) of the interactions (Figure 4A). In the TCR signaling pathway network, IFNG, LCK, CD28, CD40L, and PIK3R1 proteins from the 14 total proteins were involved in most (≥7) of the interactions (Figure 4B). Finally, in the BCR signaling pathway network, SYK, CD19, and CD79B proteins from the six total proteins were responsible for most (≥4) of the interactions (Figure 4C).

## 4. Discussion

The lymphocytes are a key component of the innate and adaptive immune systems, and thus play important roles in immunity. Among them, the γδ T cells trigger cell-mediated immune activity while fighting against foreign substances, such as viruses, microorganisms, and pathogens. The damage caused by a viral disease is severe even in the early stage of weaning swine that have acquired passive immunity [15,31]. Changes in the cell levels of CD4 and CD8, which are subgroups of receptors in γδ T cells in humans, and their ratio have been shown to be considerably inherited [32,33,34], wherein the heritability estimates for the ratios of CD8 are 65%, respectively [34]. The CD4^+^ cells are required in the priming phase for a functional CD8 memory. The CD8 antigen, a cell surface glycoprotein found in most cytotoxic T cells, mediates effective cell–cell interactions in the immune system. The CD8 antigen, along with other TCRs, can recognize the antigens presented by MHC class I antigen presenting cells in humans [31,35] as well as in pigs [10]. Thus, CD8 in the γδ T cells is characteristic of immune-related traits, and CD8^+^ in particular is involved in disease resistance and susceptibility.

In this study, we attempted to identify a precise mechanism related to the immune functions of γδ T cells in non-infected swine after sorting the swine as per their age (D10/D28) and γδ T cell types (CD8^+^/CD8^−^) and analyzing and comparing their transcriptomes using RNA-sequencing technology. First, we determined the differences in the γδ T cells as per their CD8^+^/CD8^−^ types and observed that 1207 DEGs were commonly expressed in the D10 CD8^+^ vs. D10 CD8^−^ and D28 CD8^+^ vs. D28 CD8^−^ groups, and 646 genes were differentially upregulated in the CD8^+^ samples compared to the CD8^−^ samples. Their GO and KEGG pathways were found to be related to multiple immune-related reactions, including the cytokine–cytokine receptor interaction and TCR signaling pathway, which are important in terms of immunity (Figure 2). These pathways were also validated in the GSEA (Figure 3A).

Cytokines are important intercellular modifiers and mobilizers involved in the adaptive inflammatory host defense mechanism intended for the recovery of homeostasis; growth, differentiation, and extinction of cells; and neovascularization and their development and recovery processes. The cytokine–cytokine receptor interaction is one of the most effective neurotransmitter pathways for the secretion of cytokines. A previous study on PRRSV, which causes serious damage to the productivity of swine, showed that the cytokine–cytokine receptor interaction was regulated by an immune-response-specific pathway, and that anti-viral defenses were mediated by the significant enhancement of signaling by interleukins, including cytokine–cytokine receptor interaction, BCR signaling, viral mRNA translation, IFN-gamma pathway, and AP-1 transcription factor network pathways, involving multiple genes [36,37,38]. The visualization of the expression levels of the core enriched genes in the cytokine–cytokine receptor interaction showed that 25 out of 52 genes were included in the upregulated DEG group (Figure 3B) and the analysis of the PPI network revealed that eight organically linked proteins (*CXCR3, IFNG, CCR5, TNF, CCL4, CCL5, CD40L,* and *CD27*) represented a majority (≥15) of the interactions. These were shown to be genes related to immune-related terms, such as chemokine (*CCL4, CCL5)*, chemokine receptor (*CXCR3, CCR5*), and interferon-gamma (*IFNG*; Figure 4A). These results demonstrate that CD8^+^ γδ T cells may be associated with the Th1 phenotype and play a role in cell-mediated immune responses [39].

CCR5, CCL4, and CCL5 are known suppressors of the human immunodeficiency virus (HIV) in humans. *CCL4* and *CCL5* genes are mainly expressed by the CD8^+^ T cells and have a strong relationship with inflammation [40,41]. In particular, CCR5 causes inflammation during infections, and as a co-receptor of the macrophage-tropic virus, it suppresses HIV; thus, it is likely to act in a similar manner in the case of PRRSV as well, whose primary target is the alveolar macrophage [42]. CD8^+^ γδ T cells in non-infected swine, therefore, are expected to render relatively higher immune activity against viral infections. Upregulated *CCR5*, *CCL4*, and *CCL5* in case of PRRSV infection are considered to aid in the development of immune reactions by mechanisms similar to those found in the case of an HIV infection. *CCR5* and *CCL4* genes are overexpressed in PRRSV-infected swine compared to in non-infected swine, 7 days post-infection (dpi) [43]. Recently, Lim et al. [44] reported that *CCL5*, with regard to adaptive immune signaling, is upregulated in PRRSV-infected swine at 10 dpi compared to in non-infected swine at 0 dpi.

While CXCR3 is strongly associated with the CXCL11 chemokine, CXCL11 recruits innate natural killer cells [45], is involved in protective immunity by affecting the adaptive immune reaction toward tumors and viral infections [46,47], and attracts T-effector cells [48].

Cytotoxic T lymphocyte (CTL) is a type of T lymphocyte, a leucocyte, that removes infected or damaged cells and tumor cells. The CTLs express TCR, and most of them recognize peptides of specific antigens attached to class I MHC on the surface of cells; CD8, a glucoprotein on the surface of the CTLs, is bound to a specific part of the class I MHC molecule. The binding of CD8 and the MHC molecule facilitates a strong binding between the CTL and target cell. The CD8^+^ T cell performs pre-defined cytotoxic actions in the immune system via releasing cytolytic granules [49,50,51]. Our study, through the network analysis of 14 upregulated DEGs in the TCR signaling pathway, showed that five organically linked proteins (IFNG, LCK, CD28, CD40L, and PIK3R1) represented multiple (≥7) interactions. *IFNG* and *CD40LG* were also indicated in the results of the cytokine–cytokine receptor interaction, confirming that they were included in the upregulated DEG group (Appendix A). In addition, a transcript level of perforin gene was significantly upregulated in CD8^+^ γδ T cells, indicating that this cell type may exert a similar function as CTL [52]

IFNγ, a type II interferon, is an important cytokine for innate and adaptive immune signaling; in particular, it directly inhibits viral replication in virus-infected cells [53]. CD40L, encoded by *CD40LG*, is expressed on the surface of T cells and regulates the function of cells by binding CD40 to the surface of B cells [54]. In this study, we showed that two pathways related to γδ T cell activation, TCR signaling, and cytokine–cytokine receptor interaction were more significantly enriched in CD8^+^ T cells than in CD8^−^ T cells. This is consistent with the results of a previous study on the expression of 10 genes involved in these two pathways, which showed that *IFNG* and *CD40LG* are significantly overexpressed in the CD8^+^ T cells of swine [55]. This is possibly because these genes are overexpressed in the CD8^+^ γδ T cells of swine, and thus, they affect the mechanism and interaction between the cytokine–cytokine receptor interaction and TCR signaling pathway, which are ultimately involved in passive immunity, innate immune response, and adaptive immune response in swine. The result that the TCR signaling pathway was observed more often in the CD8^+^ samples than in the CD8^−^ samples in both D10 and D28 groups suggests that the CD8^+^ γδ T cells are expected to show a relatively higher activity toward various environmental infectious factors, such as bacteria and viruses, during general breeding, where the animals are not exposed to artificial diseases.

Second, 561 DEGs were downregulated in CD8^+^ samples compared to in CD8^−^ samples, and the functional enrichment analysis based on KEGG and GO showed that both the cytokine–cytokine receptor interaction and BCR signaling pathway were the significantly enriched biological pathways associated with the immune reactions (Figure 2A,B). These results showed that there are functionally different mechanisms based on the presence of CD8^+^ on cells, although both mechanisms are related to immunity. The PPI network analysis using 22 genes, including downregulated DEGs from the core enriched genes of the BCR signaling pathway, showed that six organically linked proteins, such as SYK, CD19, and CD79B, represented the majority of the (≥4) interactions (Figure 4C). BCR has two crucial functions upon interaction with an antigen: signal transduction, involving changes in receptor oligomerization, and mediating internalization for the subsequent processing of the antigen and presentation of peptides to helper T cells [56]. Further studies are necessary in order to characterize the functional role of these genes, especially CD19 and CD79B in γδ T cells.

In addition, we compared the D10 and D28 age groups for each CD8 type (D28 CD8^−^ vs. D10 CD8^−^ and D28 CD8^+^ vs. D10 CD8^+^) and observed that CD8^−^- and CD8^+^-specific DEGs were 307 and 305, respectively, and multiple immunity-related terms were enriched in the functional analyses (Appendix A). The numbers of CD8^−^ and CD8^+^ specific DEGs were, however, smaller than the number of common DEGs (1207) based on the group comparisons of D10 CD8^+^ vs. D10 CD8^−^ and D28 CD8^+^ vs. D28 CD8^−^, suggesting that the differences based on CD8 types might be greater than those based on age. However, dynamic biological changes as per the age (D10 and D28) groups are likely to be observed in future studies on pathogenic infections (including viruses) in swine.

## 5. Conclusions

In this study, we performed DEG profiling and functional annotations using RNA-seq data to comprehend the functional differences between CD8^+^ and CD8^−^ cell types in non-infected swine. Relatively upregulated DEGs in the CD8^+^ type was functionally enriched in the cytokine–cytokine receptor interaction and TCR signaling pathway, whereas the downregulated DEGs were enriched in the BCR signaling pathway. In the cytokine–cytokine receptor interaction and TCR signaling pathway, chemokine-related genes (CXCR3, CCR5, CCL4, CCL5), interferon-gamma (IFNG), and CD40 ligand (CD40LG) were involved in the inter-regulation of functions related to immunity. Thus, this study is expected to contribute to the field by providing fundamental data to future studies on pathogenic infections, such as that by the swine virus.

## Figures and Tables

**Figure 1 animals-11-02165-f001:**
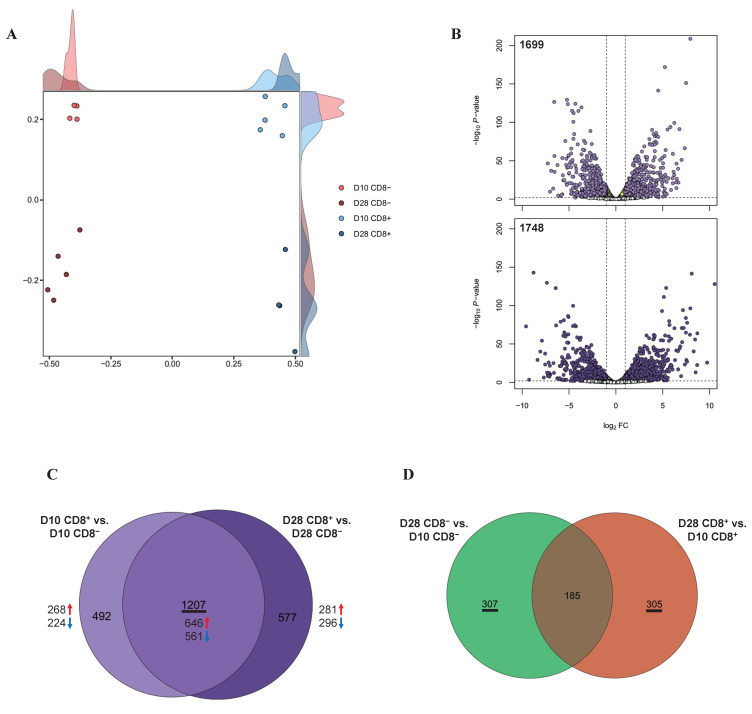
Transcriptomes in γδ T cells according to the involvement of CD8: (**A**) MDS revealed separate clusters among the four cell groups, based on the transcriptomes; (**B**) volcano plots of DEGs for CD8^+^ γδ T cells compared to CD8^−^. The x and y axes of the volcano plots show the log_2_ FCs and −log_10_
*p*-values, respectively; (**C**) Venn diagrams with DEGs from D10 CD8^+^ vs. D10 CD8^−^ (light purple) and D28 CD8^+^ vs. D28 CD8^−^ (dark purple); (**D**) Venn diagrams with DEGs from D28 CD8^−^ vs. D10 CD8^−^ (green) and D28 CD8^+^ vs. D10 CD8^+^ (orange).

**Figure 2 animals-11-02165-f002:**
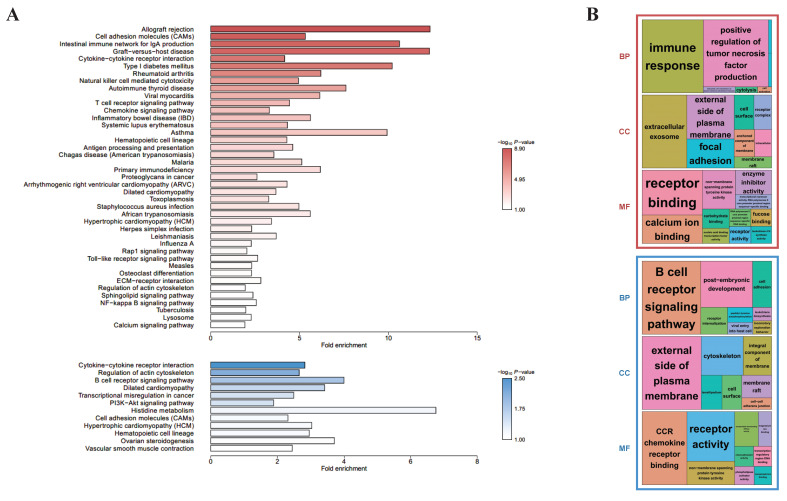
Enrichment analyses based on the DAVID database, aimed at discovering the biological meaning of common up- (red) and down-regulated genes (blue) between D10 CD8^+^ vs. D10 CD8^−^ and D28 CD8^+^ vs. D28 CD8^−^: (**A**) KEGG-enriched pathways were visualized by bar plots; (**B**) GO treemaps (biological process: BP, cellular component: CC, and molecular function: MF) were made based on the *p*-values.

**Figure 3 animals-11-02165-f003:**
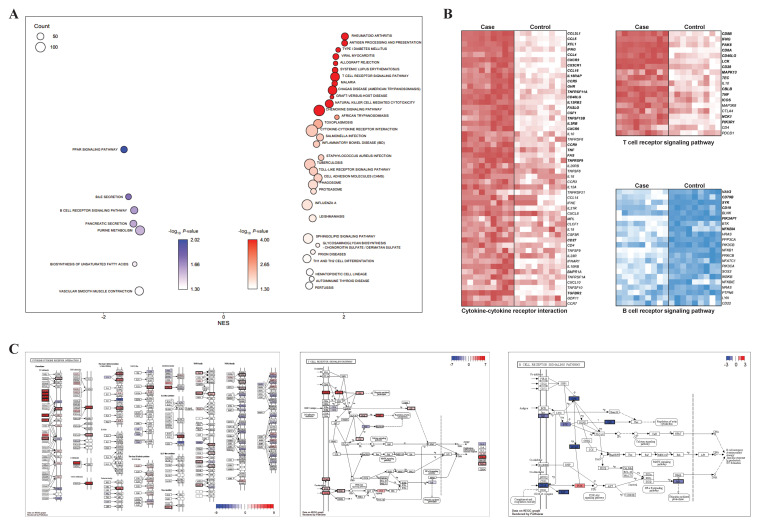
Visualized GSEA results related to three main pathways: (**A**) bubble plot for GSEA showed KEGG-enriched terms; (**B**) heatmaps for core enriched genes in three main pathways (T cell receptor signaling pathway, B cell receptor signaling pathway, and cytokine–cytokine receptor interaction) are shown. The genes in bold include common DEGs between case 1 and 2. (**C**) Pathway profiling for three main pathways were visualized with both significant genes in DEGs and GSEA. Proteins corresponding to genes are presented as log_2_ FC values and significant genes are shown with a black border.

**Figure 4 animals-11-02165-f004:**
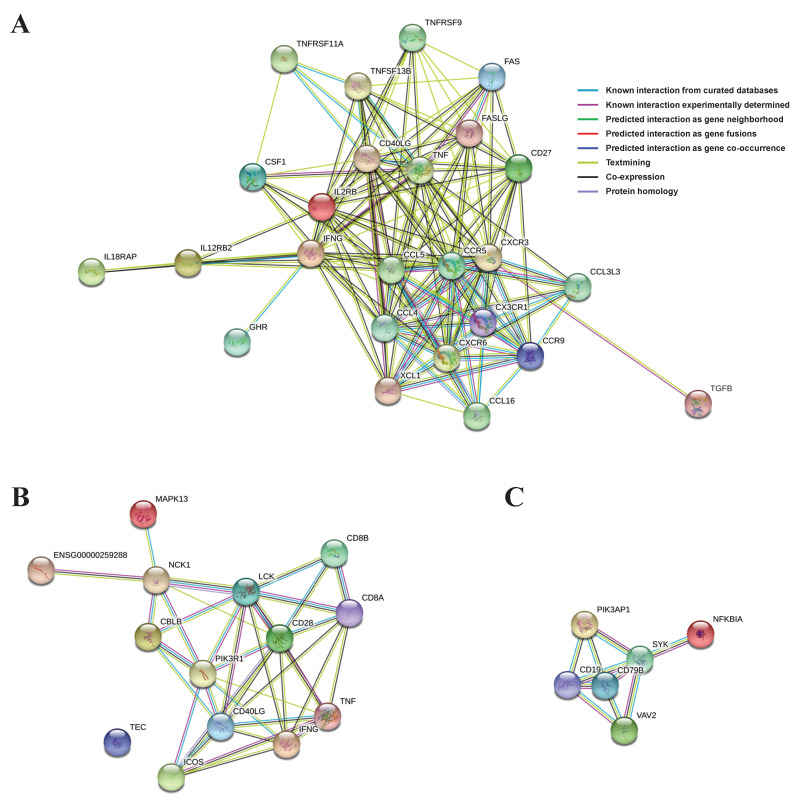
PPIs were drawn using significant genes in each main pathway: (**A**) cytokine-cytokine receptor interaction; (**B**) T cell receptor signaling pathway; (**C**) B cell receptor signaling pathway.

## Data Availability

Illumina sequencing raw reads data have been uploaded to NCBI SRA database, item number is PRJNA742154.

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
