# Peer review of "Comprehensive Transcriptomic Comparison between Porcine CD8− and CD8+ Gamma Delta T Cells Revealed Distinct Immune Phenotype"

_animals, 2021, doi:10.3390/ani11082165_

Round 1
Reviewer 1 Report
The authors addressed several of my queries in an adequate manner. However, I feel that some of my initial concerns have not been fully resolved, please see my comments on this below.
Explanation of color code:
Black: comments of this reviewer on the initial submission
Red: answers of the authors on the reviewer comments in the first round
Blue: comments of this reviewer on the answers from the authors. I ask the authors to address my comments in blue.
I completely miss a clear hypothesis why the authors focussed on CD8+ versus CD8neg GammaDeltas for their RNseq experiments. It looks like that the authors assume that CD8+ GammaDelta T cells are similar to conventional CD8 T cells? However, there are several published findings that indicate that CD8alpha expression is not immediately linked to a CD8 T- cell-like phenotype. To me it looks like that the authors ignore the finding that porcine T cells in general can express CD8alpha-alpha homodimers or CD8alpha-beta heterodimers. The current assumption is that CD8alpha-beta heterodimers are present on conventional CD8 T cells whereas CD8alpha-alpha homodimers are expressed on antigen experienced CD4 T cells, some GammaDelta T cells and the vast majority of NK cells (see cited reference No. 3). Since the authors used anti-CD8alpha antibodies for sorting I do not think there is a good basis to speculate that these cells resemble conventional CD8 T cells. Instead, the authors ignore the finding that it is rather CD2 expression that seems to separate two substantially different GammaDelta T cell subsets in pigs (DOI: 10.4049/jimmunol.1202890).
A: We agree with review’s comments on Gamma Delta T cells regarding CD2 expression. As well described in previous study (DOI: 10.4049/jimmunol.1202890), CD2 expression has been known to determine two lineages of gamma delta T cells in pigs. However, we investigated the effect of CD8 expression on gamma delta T cells by using RNA sequencing technology in this study. We observed that the frequencies of these two cell types (CD8+ gamma delta T cells, CD8- gamma delta T cells) had been changed upon viral infection, indicating a possible role of these two cell type on anti-viral immune responses (data not shown). Therefore, we decided to see if they differ in their transcriptional signatures.
Comment reviewer: What do the authors mean by “data not shown”? Are these unpublished findings? Even then I recommend mentioning them in the introduction but with some relevant details: what virus was used for infection, how was the infection done (route, dose), how and where were GammaDelta cells analyzed (flow cytometry, blood, TBLN, lung)? I still miss a clear justification in the current version of the manuscript why the authors compared CD8positive and negative GammaDelta cells. They should also address in the introduction why CD2 as another important marker to separate porcine GammaDelta T cells was ignored.
Related to this I miss a justification why the authors isolated the GammaDelta T cells from bronchial lymph nodes? Why from those lymph nodes and not from blood/ other lymph nodes/ anon-lymphoid tissue like lung or gut? The same applies to the chosen age: why 10 and 28days post weaning? Why not older or younger? Why not at weaning versus a later time point? All these important questions are not answered in the manuscript and should be addressed in the introduction.
A: Respiratory tract is one of the major route of viral or bacterial infection in pigs, causing severe respiratory diseases such PRDC. Recently, we demonstrated that local immune responses such as lung and bronchial lymph nodes could play a major role in PRRSV clearance in weaned piglets (doi.org/10.1186/s13567-020-00789-7). Based on this finding, we decide to characterize gamma delta T cells in bronchial lymph nodes initially. We are planning to expand our study further to PRRSV infected pigs.
Comment reviewer: I agree to this, but could the authors please outline this in a shortened version in the introduction and not only in the answer to me?
A: Pigs are extremely vulnerable to pathogen and viral diseases after weaning [12-14]. Accordingly, we have referred to a paper which suggests that the period with the most active activities of the immune cells are most active after being infected by a virus disease are after weaning 10-days: D10 and after weaning 28-days: D28 [15, 16], and to conduct a research on the comprehensive functions of gamma delta T cell in a non-infected state instead of a virus infection experiment, an analysis was implemented during two points (10 and 28 days). The details have been added to lines 85-90 of the introduction section.
Lines 85-90 still have exactly the same wording as in the first submission. Could the authors please state in a clearer way that this period after weaning is addressed by the sampling time points in their study, i.e. make a better connection between the statements in lines 87-90 and the chosen time points in line 94.
- Please clarify age of the animals: line 34 states animals were 10 and 28 days old. According to line 99 they were 38 and 56 days old. Lines 198-199 just state D10 versus D28 and these designations are then used throughout the manuscript.
A: We revised “at 38 days (D10) of age” to “at 38 days (after weaning 10-day: D10) of age” and “at 56 days (D28) of age” to “at 56 days (after weaning 28-day: D28) of age” following your correction.
This correction still needs to be applied to line 35 in the abstract.
- Lines 117-118: could the authors please mention how many cells of the respective subsets were sorted for subsequent RNA-isolation? I would also appreciate a supplementary figure that shows original flow cytometry data of the phenotyping and subsequent sorting purity.
A: One million cells were sorted and used for RNA sequencing. Subsequent sorting purity was 100%. Original flow cytometry data of FACSaria sorting is included in the manuscript as supplementary data (Figure S1).
Could the authors add the information about the cell numbers put into the manuscript (end of chapter 2.2) and not only mention it in the answer to me?
- Line 97 versus lines 123 and 198: According to line 97, 18 piglets were included in the study. This should have resulted in 36 RNAseq samples (CD8pos and CD8neg GammaDeltas cells for each piglet). However, later in the manuscript only data of 18 samples is presented, with an unequal distribution between age groups and phenotypes (e.g. Fig. 1A). Why were those other 18 samples omitted?
A: Thanks for your comment. We have revised the incorrect “18 piglets” correctly into “10 piglets” in all the paragraphs.
In line 124 the authors mention now that RNA was extracted from 20 samples. However, in Fig 1A and Suppl Tab 1 as well as 2+3 only the data of 18 samples is displayed. Please clarify.
- Fig. 1D, labelling of the brown pie chart, please check, this should read CD28 CD8+ vs D10 CD8+ not “D10 CD8-“.
A: We revised all the parts on which you suggested.
Sorry, I think this has not been changed and the labelling of the brown pie chart in Fig 1D is still wrong (it should be as in column “E” of supplementary table 4).
- Figures 2B and all elements of Figure 3: on an A4 format the labelling of these figures are not legible. I suggest increasing the size substantially for Fig. 2B, 3A and 3B and to move 3C into the supplement, with each pathway covering a whole A4 page.
A: Thanks for your comment. For the main figures of this paper, much effort is made for the readers so that the findings of this study can be viewed together. To ensure better editing, we revised Table 1. into Supplementary Table 3.
I agree that is good to see the findings together in one figure, but if the individual elements of that Figure are not legible (in particular Fig. 3C) this is useless.

Author Response
Comments and Suggestions for Authors
The authors addressed several of my queries in an adequate manner. However, I feel that some of my initial concerns have not been fully resolved, please see my comments on this below.
Explanation of color code:
Black: comments of this reviewer on the initial submission
Red: answers of the authors on the reviewer comments in the first round
Blue: comments of this reviewer on the answers from the authors. I ask the authors to address my comments in blue.
I completely miss a clear hypothesis why the authors focussed on CD8+ versus CD8neg GammaDeltas for their RNseq experiments. It looks like that the authors assume that CD8+ GammaDelta T cells are similar to conventional CD8 T cells? However, there are several published findings that indicate that CD8alpha expression is not immediately linked to a CD8 T- cell-like phenotype. To me it looks like that the authors ignore the finding that porcine T cells in general can express CD8alpha-alpha homodimers or CD8alpha-beta heterodimers. The current assumption is that CD8alpha-beta heterodimers are present on conventional CD8 T cells whereas CD8alpha-alpha homodimers are expressed on antigen experienced CD4 T cells, some GammaDelta T cells and the vast majority of NK cells (see cited reference No. 3). Since the authors used anti-CD8alpha antibodies for sorting I do not think there is a good basis to speculate that these cells resemble conventional CD8 T cells. Instead, the authors ignore the finding that it is rather CD2 expression that seems to separate two substantially different GammaDelta T cell subsets in pigs (DOI: 10.4049/jimmunol.1202890).
A: We agree with review’s comments on Gamma Delta T cells regarding CD2 expression. As well described in previous study (DOI: 10.4049/jimmunol.1202890), CD2 expression has been known to determine two lineages of gamma delta T cells in pigs. However, we investigated the effect of CD8 expression on gamma delta T cells by using RNA sequencing technology in this study. We observed that the frequencies of these two cell types (CD8+ gamma delta T cells, CD8- gamma delta T cells) had been changed upon viral infection, indicating a possible role of these two cell type on anti-viral immune responses (data not shown). Therefore, we decided to see if they differ in their transcriptional signatures.
Comment reviewer: What do the authors mean by “data not shown”? Are these unpublished findings? Even then I recommend mentioning them in the introduction but with some relevant details: what virus was used for infection, how was the infection done (route, dose), how and where were GammaDelta cells analyzed (flow cytometry, blood, TBLN, lung)? I still miss a clear justification in the current version of the manuscript why the authors compared CD8positive and negative GammaDelta cells. They should also address in the introduction why CD2 as another important marker to separate porcine GammaDelta T cells was ignored.
A-2: We used “data not shown” since manuscript with those results are under preparation and will be submitted to the Journal. Therefore, we are sorry that results cannot be described in this manuscript. We hope reviewer understand this situation, As reviewer mentioned, CD2 is an another important marker for porcine gdT cells but in some cases (DOI: 10.1128/jvi.01211-10), porcine gdT cells are characterized by gdTCR+ and CD8+. For this study, we missed the importance of CD2 marker but will consider to analyze transcriptome difference between with or without CD2 marker in gdT cells. We really appreciate for reviewer’s comment.
Related to this I miss a justification why the authors isolated the GammaDelta T cells from bronchial lymph nodes? Why from those lymph nodes and not from blood/ other lymph nodes/ anon-lymphoid tissue like lung or gut? The same applies to the chosen age: why 10 and 28days post weaning? Why not older or younger? Why not at weaning versus a later time point? All these important questions are not answered in the manuscript and should be addressed in the introduction.
A: Respiratory tract is one of the major route of viral or bacterial infection in pigs, causing severe respiratory diseases such PRDC. Recently, we demonstrated that local immune responses such as lung and bronchial lymph nodes could play a major role in PRRSV clearance in weaned piglets (doi.org/10.1186/s13567-020-00789-7). Based on this finding, we decide to characterize gamma delta T cells in bronchial lymph nodes initially. We are planning to expand our study further to PRRSV infected pigs.
Comment reviewer: I agree to this, but could the authors please outline this in a shortened version in the introduction and not only in the answer to me?
A-2: As reviewer suggested, we made changes as shown below.
Since the respiratory tract is one of the major routes of viral or bacterial infection, we performed RNA-sequencing (RNA-seq) analysis of the γδ T cells acquired from bronchial lymph nodes from 10 pigs that were after weaning 10 (n=5) and 28 (n=5) days.
A: Pigs are extremely vulnerable to pathogen and viral diseases after weaning [12-14]. Accordingly, we have referred to a paper which suggests that the period with the most active activities of the immune cells are most active after being infected by a virus disease are after weaning 10-days: D10 and after weaning 28-days: D28 [15, 16], and to conduct a research on the comprehensive functions of gamma delta T cell in a non-infected state instead of a virus infection experiment, an analysis was implemented during two points (10 and 28 days). The details have been added to lines 85-90 of the introduction section.
Lines 85-90 still have exactly the same wording as in the first submission. Could the authors please state in a clearer way that this period after weaning is addressed by the sampling time points in their study, i.e. make a better connection between the statements in lines 87-90 and the chosen time points in line 94.
A-2: We have added information for the early stage of weaning to have better connection as the reviewer suggested (page 2, lines 91 & 98~99).
- Please clarify age of the animals: line 34 states animals were 10 and 28 days old. According to line 99 they were 38 and 56 days old. Lines 198-199 just state D10 versus D28 and these designations are then used throughout the manuscript.
A: We revised “at 38 days (D10) of age” to “at 38 days (after weaning 10-day: D10) of age” and “at 56 days (D28) of age” to “at 56 days (after weaning 28-day: D28) of age” following your correction.
This correction still needs to be applied to line 35 in the abstract.
A-2: As the reviewer commented, we have revised “10-day-old (D10) and 28-day-old (D28)” to “38-day-old (after weaning 10-day: D10) and 56-day-old (after weaning 28-day: D28)” (page 1, lines 35-36).
- Lines 117-118: could the authors please mention how many cells of the respective subsets were sorted for subsequent RNA-isolation? I would also appreciate a supplementary figure that shows original flow cytometry data of the phenotyping and subsequent sorting purity.
A: One million cells were sorted and used for RNA sequencing. Subsequent sorting purity was 100%. Original flow cytometry data of FACSaria sorting is included in the manuscript as supplementary data (Figure S1).
Could the authors add the information about the cell numbers put into the manuscript (end of chapter 2.2) and not only mention it in the answer to me?
A-2: As reviewer suggested, following sentences were added into introduction.
Finally, one million cells were stored in liquid nitrogen and subsequently used for RNA-seq analysis.
- Line 97 versus lines 123 and 198: According to line 97, 18 piglets were included in the study. This should have resulted in 36 RNAseq samples (CD8pos and CD8neg GammaDeltas cells for each piglet). However, later in the manuscript only data of 18 samples is presented, with an unequal distribution between age groups and phenotypes (e.g. Fig. 1A). Why were those other 18 samples omitted?
A: Thanks for your comment. We have revised the incorrect “18 piglets” correctly into “10 piglets” in all the paragraphs.
In line 124 the authors mention now that RNA was extracted from 20 samples. However, in Fig 1A and Suppl Tab 1 as well as 2+3 only the data of 18 samples is displayed. Please clarify.
A-2: As we added “Further analyses were performed after filtering outliner based on MDS result.” in previous revision (page 4, lines 179-180), further information have been described to clearly mention the materials and methods (page 5, lines 221-222).
- Fig. 1D, labelling of the brown pie chart, please check, this should read CD28 CD8+ vs D10 CD8+ not “D10 CD8-“.
A: We revised all the parts on which you suggested.
Sorry, I think this has not been changed and the labelling of the brown pie chart in Fig 1D is still wrong (it should be as in column “E” of supplementary table 4).
A-2: We have revised the Fig 1D and separately attached high resolution file.
- Figures 2B and all elements of Figure 3: on an A4 format the labelling of these figures are not legible. I suggest increasing the size substantially for Fig. 2B, 3A and 3B and to move 3C into the supplement, with each pathway covering a whole A4 page.
A: Thanks for your comment. For the main figures of this paper, much effort is made for the readers so that the findings of this study can be viewed together. To ensure better editing, we revised Table 1. into Supplementary Table 3.
I agree that is good to see the findings together in one figure, but if the individual elements of that Figure are not legible (in particular Fig. 3C) this is useless.
A-2: As the reviewer suggested, we have substantially increased the size for Fig 2B, 3A, and 3B. In addition, Fig 3C was moved to Fig S3, S4, and S5.

Reviewer 2 Report
There are no further comments.
Author Response
Thanks for your comments
This manuscript is a resubmission of an earlier submission. The following is a list of the peer review reports and author responses from that submission.
Round 1
Reviewer 1 Report
The manuscript by Kim et al. reports on the transcriptome identified in CD8+ and CD8negative GammaDelta T cells, isolated from the lymph nodes of young pigs. Following cell sorting of the aforementioned GammaDelta subsets, the authors applied RNAseq. Differentially expressed genes are described and analysed by bioinformatic tools for their functional relevance.
This kind of analysis for porcine GammaDelta T-cell subsets is novel and the performed experiments seem to be of adequate technical quality. However, I suggest numerous improvements on the way the data is presented and the statements and conclusions in the manuscript.
I completely miss a clear hypothesis why the authors focussed on CD8+ versus CD8neg GammaDeltas for their RNseq experiments. It looks like that the authors assume that CD8+ GammaDelta T cells are similar to conventional CD8 T cells? However, there are several published findings that indicate that CD8alpha expression is not immediately linked to a CD8 T-cell-like phenotype. To me it looks like that the authors ignore the finding that porcine T cells in general can express CD8alpha-alpha homodimers or CD8alpha-beta heterodimers. The current assumption is that CD8alpha-beta heterodimers are present on conventional CD8 T cells whereas CD8alpha-alpha homodimers are expressed on antigen experienced CD4 T cells, some GammaDelta T cells and the vast majority of NK cells (see cited reference No. 3). Since the authors used anti-CD8alpha antibodies for sorting I do not think there is a good basis to speculate that these cells resemble conventional CD8 T cells. Instead, the authors ignore the finding that it is rather CD2 expression that seems to separate two substantially different GammaDelta T cell subsets in pigs (DOI: 10.4049/jimmunol.1202890).
Related to this I miss a justification why the authors isolated the GammaDelta T cells from bronchial lymph nodes? Why from those lymph nodes and not from blood/ other lymph nodes/ a non-lymphoid tissue like lung or gut? The same applies to the chosen age: why 10 and 28days post weaning? Why not older or younger? Why not at weaning versus a later time point? All these important questions are not answered in the manuscript and should be addressed in the introduction.
Further points:
I recommend showing the DEGs presented in Fig 1B also as supplementary tables, listing individual genes with their names, logFC and p-values, as well as TPM values. Authors should also consider submitting their sequencing data to an online repository, like Gene Expression Omnibus.
Please clarify age of the animals: line 34 states animals were 10 and 28 days old. According to line 99 they were 38 and 56 days old. Lines 198-199 just state D10 versus D28 and these designations are then used throughout the manuscript.
Line 79: not all pigs have >70% GammaDelta T cells in the blood. Check reference No. 3 (Table 1)
Lines 117-118: could the authors please mention how many cells of the respective subsets were sorted for subsequent RNA-isolation? I would also appreciate a supplementary figure that shows original flow cytometry data of the phenotyping and subsequent sorting purity.
Line 97 versus lines 123 and 198: According to line 97, 18 piglets were included in the study. This should have resulted in 36 RNAseq samples (CD8pos and CD8neg GammaDeltas cells for each piglet). However, later in the manuscript only data of 18 samples is presented, with an unequal distribution between age groups and phenotypes (e.g. Fig. 1A). Why were those other 18 samples omitted?
Line 137: which kit was used to prepare the cDNA libraries?
Line 143: “…resulting in an average of 63.3 million reads per sample.” However, in Suppl. Table 1, between 110 and 140 million reads per sample are listed. Please clarify.
Line 205: typo, “MDS”, not “MSD”.
Lines 210-211: Table 1 shows significant DEGs, not false discovery rate. Please check.
Fig. 1D, labelling of the brown pie chart, please check, this should read CD28 CD8+ vs D10 CD8+ not “D10 CD8-“.
Line 230: please do not abbreviate “cytokine-cytokine receptor” with “CCR”. This abbreviation is widely used in immunology for “chemokine receptor” and hence it is highly confusing to have the same abbreviation in this manuscript but with a different meaning.
Figures 2B and all elements of Figure 3: on an A4 format the labelling of these figures are not legible. I suggest increasing the size substantially for Fig. 2B, 3A and 3B and to move 3C into the supplement, with each pathway covering a whole A4 page.
Comments on the discussion
Overall, the discussion would benefit from some clearer statements what the identified differences between CD8positive and negative GammaDelta T cells could mean. The presence of CCR5, CXCR3 and IFN-gamma transcripts suggest some differentiation towards a Th1 or type-1 phenotype, an aspect that the authors do not consider (apart from speculating about a role in viral infections, lines 344-345). And when they come back to their initial speculation about a cytotoxic activity of these cells (lines 356 to 365) they do not consider transcripts involved in cytolytic functions. Perforin expression in porcine CD8alpha+ GammaDelta cells has been shown on the protein level, for example in doi: 10.3389/fimmu.2019.00396. So, what was the transcription level for perforin in this study? What about other molecules involved in cytolytic activity, like granzymes?
Line 303-304: “Changes in the cell levels of CD4 and CD8, which are subgroups of receptors in γδ T cells,….” I am not aware of published data on CD4 expressing GammaDelta T cells in pigs. Please clarify or rephrase. The same applies to the sentence in lines 310-312.
Lines 370 to 372: please mention that this statement refers to GammaDelta T cells, not just T cells.
Lines 383 to 399: I disagree with some of the statements in this paragraph. On what basis do the authors conclude that CD8neg GammaDelta T cells may communicate with B cells? I cannot follow this argument. Moreover, I do not see the relevance of the statement on the lack of monoclonal antibodies for porcine B cell markers. Total B cells can be identified in pigs for example with an anti-CD79alpha mAb and no CD79alpha expressing GammaDelta T cells can be identified with this antibody (see for example doi: 10.1016/j.dci.2013.01.003).
Simple Summary and Abstract (lines 30-31 and 43-44, respectively): I consider the statements that the data presented in this manuscript is “fundamental” or “primary” as an overemphasis of what is provided. Please rephrase, e.g. something like “contributes to our understanding of porcine GammaDelta T cell biology.
Reviewer 2 Report
Based on the reviewer’s evaluation, the work entitled “Comprehensive transcriptomic comparison between porcine CD8- and CD8+ gamma delta T cells revealed distinct immune phenotype” is considered to be suitable for publication in Animals. Before acceptance, some comments should be considered.
- Why use D10 and D28 time-points, the reason should be given.
- Besides the common upregulated and downregulated DEGs, the time-point specific upregulated and downregulated DEGs (for developmental concerns) may be more interesting for some readers. Suggest adding them into the manuscript.
- Lines 160-162, in total, there are 6 comparisons, why use these 4 ones?
- The results of flow cytometry should be given in supplementary materials.
- In general, the differentially expressed genes should be partially selected to be validated by qRT-PCR or similar techniques.